# The Effect of the Energy Release Rate on the Local Damage Evolution in TRIP Steel Composite Reinforced with Zirconia Particles

**DOI:** 10.3390/ma16010134

**Published:** 2022-12-23

**Authors:** Shao-Chen Tseng, Chen-Chun Chiu, Faisal Qayyum, Sergey Guk, Ching-Kong Chao, Ulrich Prahl

**Affiliations:** 1Institut für Metallformung, Technische Universität Bergakademie Freiberg, 09599 Freiberg, Germany; 2Department of Mechanical Engineering, National Taiwan University of Science and Technology, Taipei 106335, Taiwan

**Keywords:** ceramic, TRIP steel, in situ test, crystal plasticity, damage, DAMASK, ceramic/matrix interface, damage model

## Abstract

In this study, the effect of the energy release rate on the transformation-induced plasticity (TRIP) steel composite reinforced with 5 vol% ceramic particles is determined using the crystal plasticity simulation of the coupled brittle-ductile damage model and validated by experimental results. A miniature dog bone tensile sample is subjected to an interrupted in situ quasi-static tensile test up to a true strain of 20.3%. Using the commercial digital image correlation program VEDDAC and the image processing method in MATLAB, the test data are utilized to monitor the progress of local microstrain and damage. The impact of the energy release rate of ceramic particles is investigated by simulation using a coupled crystal plasticity-dislocation density model with ductile–brittle criteria for the corresponding phases. It can be shown that the local deformations predicted by the numerical simulation and the experimental data are qualitatively comparable. The damage pixel of the experiment, smaller E_cr_ (1.0 × 10^8^), and larger E_cr_ (1.2 × 10^8^) cases of energy release rates are 4.9%, 4.3%, and 5.1%, respectively. Furthermore, on a global strain of 20.3%, the relative error between simulation and experimental validation of smaller E_cr_ (1.0 × 10^8^) and larger E_cr_ (1.2 × 10^8^) cases is 12.2% and 4%, respectively.

## 1. Introduction

In recent years, zirconia has become one of the most significant ceramic materials. In terms of microstructure, zirconia has three well-established polymorphs: cubic, tetragonal, and monoclinic phases [1]. The cubic phase crystallizes at 2680 °C in the cooling stage. Furthermore, the intermediate tetragonal phase is formed at 2370 °C. Finally, the martensitic phase transformation (from tetragonal to monoclinic) occurs at 1047 °C [2]. This temperature-induced transformation caused a 3–5% volume expansion and probably caused cracking in a pure zirconia ceramic material [3,4]. Therefore, to prevent the damage induced by volume expansion, stabilized oxides, such as CaO, MgO, CeO_2,_ and Y_2_O_3_, were added to pure zirconia to generate the alloy materials, i.e., yttria-stabilized zirconia (YSZ) and partially stabilized zirconia (PSZ) [5]. When compared to other structural ceramics (Al_2_O_3_ and Si_3_N_4_), the thermal properties of PSZ, such as thermal conductivity and coefficient of thermal expansion, exhibited remarkable performance and phase relationships [6]. To avoid a thermal expansion mismatch between the metallic and nonmetallic materials, the thermal expansion of PSZ was moderately high, which was suitable for the reinforcement of the metal matrix composite with the nonmetallic material. Therefore, 7 to 8 mol% of YSZ was applied as a topcoat in thermal barrier coatings (TBC) to withstand the high-temperature exposure due to its low conductivity and a relatively high coefficient of thermal expansion [7]. Furthermore, magnesia-partially stabilized zirconia (Mg-PSZ) was chosen as the nonmetallic reinforcement in the metastable austenite composite due to its high strength and toughness [8,9].

Unlike other structural ceramics, zirconia ceramics feature a unique toughening mechanism, called transformation toughening, that provides superior mechanical properties, such as strength and fracture toughness [6,10]. The stress field around the crack was able to toughen the tetragonal phase transformation because it provided the requisite shear-strain component to activate the martensitic transformation [11], known as the stress-induced martensitic transformation. Furthermore, the volume expansion of the transformed ceramic particles around the tip of the crack suppressed crack propagation. With this stress-induced transformation, zirconia ceramic particles played a vital role in the metastable austenitic steel composite with Mg-PSZ particles, which indicated that samples with 5 vol-% zirconia presented superior mechanical behavior when compared to zirconia-free samples and composites with a higher zirconia content [12]. In addition to martensitic transformation in zirconia, martensitic phase transformations were observed in the steel matrix (austenitic phase to martensite phase) [13,14,15,16,17,18]. Specifically, the austenite-to-martensite transformation involved intersections of shear bands consisting of overlapping stacking faults, ε-martensite, and mechanical twins [19,20]. The volume fraction of $\alpha^{\prime }$-martensite increased when that of ε-martensite reached its maximum and subsequently decreased. Therefore, according to the transformation sequence, the strain-induced process transforms the γ-austenite to $\alpha^{\prime }$-martensite (bcc) via ε-martensite (hcp) [21,22]. Due to external loading, strain-induced martensitic transformation in the matrix occurs in the neck region, which prolongs elongations, restricting the dislocation movement around the martensitic island and the ceramic/matrix interface [23].

The individual analysis of experimental [13,24,25] and simulation [14,23,26] methods revealed that more research is required on the effects of damage on the matrix, ceramic particles, and interface of austenitic TRIP steel with particle-reinforced composite. Weidner et al. [13] observed two major damage behaviors of zirconia particles in the in situ tensile tests. The SEM images specifically showed interfacial delamination and brittle ceramic damage at different stress levels. However, few experimental articles have quantitatively evaluated the damage to the ceramic particles, the interface, and the matrix to compare with simulation analyses. For simulation analysis, a constitutive mathematical material model was used to simulate the representative volume elements (RVE) using the Düsseldorf Advanced Material Simulation Kit (DAMASK) [27], which calculates local deformation, martensitic transformation, and mechanical twinning of the austenitic stainless matrix as a consequence of the quasi-static tensile test. Compared to the experimental flow curve, the simulation model was validated under the assumption of an elastic-plastic matrix and elastic particles [14,26]. To eliminate the overestimation condition in comparison to a natural state, damages within different phases must be considered, and this can be solved by a robust phase field crystal plasticity-based model in DAMASK [28]. Furthermore, different damage criteria must be applied individually for different phases [23], which implies that the damage criteria of the zirconia particles and matrix are critical strain energy and plastic strain, respectively. However, few researchers have qualitatively and quantitatively analyzed the impact of the strain energy release rate of zirconia ceramic particles within the TRIP steel on the underlying deformation and damage behavior using experiments, simulation, and comparison via the image processing function. 

In this article, a simulation for parameter analysis and experimental validation is proposed to understand the deformation and the impact of the energy release rate of ceramic particles on the damage analysis of the Cr-Mn X3–16–7–6 Cr-Mn-Ni steel compound (TRIP steel) reinforced by Mg-PSZ particles. An in situ SEM tensile test is performed to investigate the microstructure deformation. The crystal plasticity simulation (DAMASK) is considered for analyzing realistic conditions such as material behavior, geometry, and loading. The methodology is presented in Section 2, which includes the flow chart, experiments, and simulation. The global and local results are presented in Section 3. Additionally, qualitative and quantitative analyses are used to evaluate the comparative data between the experiment and the simulation. The discussion and conclusion are presented in Section 4 and Section 5, respectively.

## 2. Methodology

In this article, the impact of the energy release rate of the ceramic particles within austenitic steel is investigated through simulation parameter analysis and experimental validation. To clarify the entire process of the present study, a flow chart with two different procedures is displayed in Figure 1.

Before the in situ SEM tensile loading test is conducted, the tensile sample of the ceramic particles and the austenite steel composite is molded into a miniature dog bone sample from the sintered composite disc using a water jet. The global stress–strain curve is determined by the tensile test [29]. The SEM image of the microstructure, which includes ceramic and austenite, was recorded under specific strain conditions. The local strain distribution and damage evolution are determined using DIC (VEDDAC) and the image processing method (MATLAB), respectively. A Python program is used to construct the interfacial geometry points from the realistic original micrograph to the simulation geometry. For the simulation model, which is based on a coupled crystal plasticity-dislocation density model for the combined ductile–brittle mode, the critical strain energy and plastic strain are implemented for damage models of the ceramic and matrix, respectively. The parameters in the simulation are validated using the experimental global stress–strain curve. Furthermore, the simulation provides a comparable local deformation and predicts the stress, dislocation density, and damage. The damage behavior is simulated by different energy release rates of the ceramic particles and validated by experimental observations. 

### 2.1. Experimentation

In this article, in situ tensile tests were performed on specially prepared specimens during deformation inside the SEM chamber. The miniature dog-bone-shaped sample is made of X3–16–7–6 Cr-Mn-Ni steel and 5 vol% Mg-PSZ composite. During the in situ SEM tensile testing, the global behavior of the true stress–strain curve and the SEM pictures were assessed [29]. The DIC method was primarily used to investigate the local von Mises strain distribution. The entire methodology of the in situ tensile tests, specimen, and DIC operation is detailed in this section.

The hot-pressured sintering technique was used to manufacture the investigated material from two components. Gas-atomized steel powder has an austenitic structure. To manufacture the composite sample, MgO-partially stabilized ZrO2 (Mg-PSZ) ceramic powder was introduced and mixed with steel powder before sintering. Metastable high-alloyed Cr-Mn-Ni cast steel is a significant component of austenite steel (X3CrMnNi16–7–6). As the reinforced component, 5% Mg-PSZ powder was chosen. Table 1 displays the chemical composition of the metastable high-alloyed Cr-Mn-Ni steel and the ZrO2 ceramic. The average grain size of zirconia ceramic exceeds 50 µm.

The in situ investigation was conducted using the observation based on previously published work [29]. As shown in Figure 2a,b, the in situ tensile testing was performed with a ZEISS Gemini SEM 450 scanning electron microscope using a Kammrath and Weiss tensile test stage. For tensile testing, the dog-bone-shaped specimen was placed on a screw-driven loading device at a forming speed of 5 μm/s at room temperature. The interrupted in situ tensile tests were performed up to a true strain of 20.3%.

As illustrated in Figure 2c, a dog-bone-shaped sample is chosen as the in-situ sample; it has a total length of 42 mm, a gauge length of 10 mm, a cross-section of 4 mm^2^, and a thickness of 1 mm. The sample used in this study has a smaller gauge length and lower thickness, resulting in a different total elongation and hardening behavior than samples with a larger gauge length and thickness. To avoid material waste and residual stress concentration along the cutting surface, the sample is cut from cylindrical discs with a diameter of 148 mm and a height of 33 mm using the water jet cutting method. Before the tensile test, the specimen was polished with sandpaper, with sizes ranging from 46.2 to 1 µm. A series of images was captured during the tensile test, and the resolution was defined as 2048 × 1536 pixels. The acceleration voltage and working distance were selected as 20 kV and 17.22 mm, respectively.

The austenite matrix is prone to larger elongation and severe damage during in situ tensile tests because of its ductile characteristics. Therefore, the deformation and damage behavior are determined by the austenite region. In this study, a noncontact method called DIC is adopted to determine the displacement field induced by tensile loading. The SEM image is initially evaluated in two-dimensional coordinates. All reference points for the DIC analysis [30] are determined by the original location in the reference field (n × n pixels) as a black dot for the initial image. Then, the new location of the measuring point is found as a red dot in the measured field (m × m pixels) for the new image. The DIC two-dimensional system is adopted by VEDDAC from Chemnitzer Werkstoffmechanik GmbH. The optimal grid size in the austenite matrix is 20 × 20 pixels. In this article, the reference and measure fields are defined as 30 × 30 and 90 × 90 pixels, respectively.

### 2.2. Simulation Method

#### 2.2.1. Boundary Condition with DAMASK Simulation

To solve the formulation of continuum mechanics, the spectral method using the fast Fourier transform [31] is used. A uniaxial monotonic tensile load is defined as follows:(1)$${\dot{F}}_{ij}=\left[\begin{array}{ccc} 1 & 0 & 0\\ 0 & * & 0\\ 0 & 0 & * \end{array}\right]\times 10^{-3}\cdot s^{-1}$$
(2)$$P_{ij}=\left[\begin{array}{ccc} {}* & * & *\\ {}* & 0 & *\\ {}* & * & 0 \end{array}\right]Pa$$
where ${\dot{F}}_{ij}$ represents the coefficients of the macroscopic rate of the deformation gradient, $P_{ij}$ is the First Piola–Kirchhoff stress. ${\dot{F}}_{11}=1$ indicates tensile condition, 0 is represented as restricted, and * is an arbitrary value during the simulation. The strain rate of all simulations is assumed to be $1\times 10^{-3}/s$ in conjunction with the periodic boundary conditions in all directions.

Regarding the construction geometry of the TRIP steel with Mg-PSZ ceramic, the TRIP steel, and the ceramic particles, can be distinguished by their different grayscale values. Figure 3a shows the initial digital SEM image. As seen in Figure 3b, the ceramic particles and the austenitic matrix are marked in white and black, respectively, to precisely identify the different microstructures. As illustrated in Figure 3c, the two phases of the microstructure, which include the ceramic particles (red) and the austenite matrix (blue), can be identified by Python coding and displayed by ParaView. The initial grain orientation must be defined at the beginning of the simulation. However, this investigation did not consider electron backscatter diffraction (EBSD). Consequently, the three Euler angles were initially set to 0. The Python scripts were applied to collect both global and local data for post-processing. Furthermore, graphical forms were exported using ParaView to display the visual figures.

#### 2.2.2. Dislocation-Based Model and Damage Criteria for Crystal Simulation

The simulation is implemented by DAMASK, which includes a couple of dislocation density-based crystal plasticity models and a phase-field damage model. In this article, the complicated viscoplastic behavior of austenite steel [32,33] is described using the dislocation density crystal plasticity model. In Appendix A and Appendix B, respectively, the constitutive laws of the dislocation-based model for crystal simulation [34,35,36] and damage criteria [37,38,39,40] are discussed. 

After phase transformation, both ε-martensite and $\alpha^{\prime }$-martensite are treated as single-phase materials to simplify the model complexity. The ceramic particles are described as elastic via tensile loading. Table 2 displays the twinning, transformation, and dislocation slip parameters for this material behavior. Table 3 displays the elastic properties of austenite, the transformed martensite phase, and the ceramic particles. To analyze the damage behavior caused by the impact of the energy release rate of the ceramic particles, the critical energy release rate is assumed to be 1.0 × 10^8^ and 1.2 × 10^8^ Jm^−2^. Table 4 and Table 5 present all the parameters and values of brittle and ductile damage for the ceramic particles and matrix, respectively. Previous research [38] has described the phase field model and the plasticity crystal model.

## 3. Results

### 3.1. Global Behavior

Figure 4 depicts the comparison between the simulation and the experimental observations at different global true strains. The simulation and experimental results are indicated by the solid and dashed lines, respectively. The red, black, blue, and green solid lines represent the stress, evolution of damage, phase transformation fraction, and total dislocation density on a logarithmic scale, respectively. The corresponding SEM images at global strains of 0.4%, 3%, 9%, 15.6%, and 20.3% are displayed on the top side of Figure 4.

As observed, the global stress for both simulation and experiment is consistent, which confirms the accuracy assumption of the crystal plasticity parameters. From the stress–strain curve, the elastic region is displayed at a global strain of 0.4%. The plastic part is displayed at global strains of 3%, 9%, 15.6%, and 20.3%. It is observed that for the simulation model, the transformation function (blue line) initiates at nearly 5% and exponentially increases up to 20% true strain. Similarly, the total dislocation density (green line) displays an increasing parabolic relationship, up to 10% of the true strain, on a logarithmic scale. Smaller deformation initially restricts transformation and dislocation density. Furthermore, when the plastic strain is up to 9% global strain, the apparent global transformation fraction is more visible. The global damage (black line) displays an exponential decrease that exceeds 9% of the global strain, which implies that the damage situation is observable. Interfacial damage can be identified at 9% global strain by experimental observation. The local result must be considered for better visualization analysis to understand the evolution of the microstructure behavior.

### 3.2. Local Result

As evidenced by the global results of the validation between experiment and simulation, a wider range of local material behavior can be predicted using numerical simulation with realistic microstructure and material parameters. Figure 5 (strain range of 0 to 40%) displays the evolution of local strain through simulation and experiment at global strains of 3%, 9%, 15.6%, and 20.3%. Austenite exhibits distinct deformation in the simulation and experimental analyses due to its ductile characteristics. Therefore, ParaView thresholds out the ceramic particles and displays only the austenite matrix. Moreover, the ceramic particles are ignored during the DIC (VEDDAC) process. The smaller plastic deformation causes the local deformation of the simulation and experiment to display a small value at 3% global deformation. In the simulation result, the strain concentration can be found near the particle corner. Although a slight difference exists between the simulation and the experiment, there can be several similar regions between them, which is indicative of the more considerable strain occurring in the middle region and corners of the particles. Section 4 details the primary reason for the slightly different strain distribution between the simulation and the experiment.

Figure 6 displays the simulation predictions of the local distribution of the dislocation density, phase transformation, and von Mises stress. The higher strain in Figure 5 is observed to correspond well with the higher dislocation density in Figure 6, which indicates that the accumulated dislocation slip is contributed by larger plastic deformation. Meanwhile, it is found that the transformation from austenite to martensite occurs at locations where the driving strain is large or near the ceramic/matrix interface. Later, the transformation region restricts the sliding and climbing motions of the dislocation. Moreover, lower strain and higher stress are observed in this transformation region. The stress is considerably influenced by the heterogeneous material as a consequence of the continued displacement and the resultant force along the interface between austenite and ceramic particles. The stress exhibits a more pronounced concentration effect as well as a strengthening of martensitic transformation and dislocation density with increasing global strain. As the dislocation density is increased, crack initiation and propagation are likely to occur as a consequence of exceeding the critical plastic strain for the ductile material (austenite) and the strain energy for the brittle material (ceramic particles).

### 3.3. Damage Behavior

#### 3.3.1. Damage Behavior from Experiment

Figure 7 depicts the qualitative and quantitative analysis of the evolution of damage via experiments employing different global strains and different approaches. In Figure 7, the damage pixel on the *y*-axis is normalized by the total pixel for each micrograph. The ceramic particle in the upper right is not considered since it vanishes in the micrograph at a global deformation of 20%. As shown on the right-hand side of Figure 7a–d, the detection process is divided into four steps, which include threshold [47], filtering [48], free-hand regions of interest (ROI) [49], and flood fill [50]. An example of the 9% global strain is used to present the qualitative results of the four mentioned steps. As illustrated in Figure 7a, the micrograph is initially detected via a grayscale threshold value (from 0 to 255) [47]. In this study, the threshold value is assumed to be 30, which implies that a grayscale value below 30 will be marked. As illustrated, the damaged region around the ceramic/matrix region and the error-detected pixel in the matrix region are both included in the damage pixel. In the second step, the entire micrograph pixel is filtered by a physical quantity, “solidity,” which is defined as the area fraction of the region compared to its convex hull [48], to eliminate the error-detected pixel in the matrix. Solidity, which is used to filter the matrix, is assumed to be in the range of 0–0.8 in this study and could eliminate the circular-shaped hole (above 0.8), as shown in Figure 7b. In the third step, as displayed in Figure 7c, the rest of the error-detected pixel is easily deduced by the free-hand region of interest (ROI) function [49]. Thus, in the third step, all the error-detected pixels in the matrix are eliminated. In the final step, the flood fill function correctly detected the damaged region adjacent to the ceramic particles [50], as shown in Figure 7d. Since the grayscale is greater than 30, the error-detected pixel adjacent to the interface is observed at the initial stage. Therefore, the damage pixel in the fourth step is normalized by the damage pixel in the initial stage. The fourth step is observed to be the correct method for detecting the damage pixel in the microstructure.

The corresponding relationship between the damage pixel and the global strain can be obtained by these four steps when the micrograph is loaded into the MATLAB Image Segmenter. The quantitative damage pixel with different global strains using the four-step detection process is illustrated on the left-hand side of Figure 7. In the first step, owing to several error-detected pixels, the damage pixel is inaccurate when the global deformation increases. Furthermore, the damage pixel in the second step is lower than that in the first step. The damage pixel in the third step is slightly altered because some error-detected pixels are detected. Moreover, in this step, the damage pixel is intensified with larger global strains. The final fourth step is represented as a red dashed line. The damage pixel between the third and fourth steps is identical at the lower strain level due to their elastic behavior. However, at a global strain of 20.3%, the damage pixel of the fourth step is 1.4% higher than that of the third step. In the third step, the severely damaged region is not entirely identified, and only a portion of the true pixel is calculated. It is inferred that the third step is significantly underestimated when the global strain is larger than the fourth step. The damage pixel in the fourth step can be regarded as a more realistic condition. Particularly, two severe damage intervals are detected in the ranges of 2.8–9% and 15.6–20% global strain.

#### 3.3.2. Damage Results from Simulation

Figure 8 presents the qualitative and quantitative damage behavior of the simulation at 9%, 15.6%, and 20.3% of the global strain. The two different energy release rates (E_cr_) for the ceramic particles for the parameter analysis are E_cr_ = 1.0 × 10^8^ Jm^−2^ and E_cr_ = 1.2 × 10^8^ Jm^−2^. The qualitative results of the smaller E_cr_ (1.0 × 10^8^ Jm^−2^) are displayed on the left-hand side of the damage image and the dashed line of the damage pixel. The qualitative results of the larger E_cr_ (1.2 × 10^8^ Jm^−2^) are displayed on the right-hand side of the damage image and a solid line of the damage pixel. In the qualitative damage figure, the gray color indicates damage, the red color indicates the ceramic particles, and the blue color indicates the austenite matrix. The damage pixel of the austenite matrix and the ceramic particles are represented by red and black colors, respectively, for the quantitative result.

The microstructure of the plastic scenario at 9% exhibits almost safe behavior for two different energy release rate models. The ductile damage is observed as the strain rises to 15.6% and increases by 45 degrees to the loading direction. Cracks form in the high-deformation region due to the higher critical plastic strain in the austenite matrix. As shown in Figure 5 and Figure 6, the evolution region of damage is consistent with the high local deformation and dislocation density region. Furthermore, the ceramic particle in the bottom left with a smaller value of E_cr_ has brittle damage due to exceeding the critical energy and is marked by a red dashed frame. More hazardous brittle damage in the ceramic region and ductile damage in the austenite region can be observed when the global strain is increased to 20.3%. The material degradation is speculated to be attributed to the accumulation of the multitude of dislocations induced by intense deformation. Crack coalescence is observed between the brittle ceramic and ductile austenite regions in a smaller E_cr_ case at 20.3% global strain for the bottom left ceramic particle, marked by a red dashed frame.

Regarding the quantitative results for damage behavior, damage pixels of both austenite and ceramic can be obtained by simulation. The damage pixel of austenite or ceramic is normalized by the total pixel of the individual material on the micrograph. The damage pixel increases as the global strain increases, as expected. The damage pixel is zero for both phases before the global strain of 9%. The increasing behavior of damage pixels can be observed for 9% to 15.6% of global strains. A steeper line is observed until the global strain reaches 20.3%. In these simulation results, the occurrence of severe damage at 15.6% global strain can be attributed to the corresponding damaged area reaching the critical value. Consequently, the high strength of the composite drastically deteriorates. As expected, the destruction level of the ceramic particles in the smaller E_cr_ (1.0 × 10^8^) case is higher than that in the larger E_cr_ (1.2 × 10^8^) case. The smaller E_cr_ (1.0 × 10^8^) case at 15.6% global strain makes it easier to attain the critical driving force in the ceramic particles, initiating the brittle crack. At a global strain of 20.3%, the damage pixel of the ceramic particles with an E_cr_ value of 1.0 × 10^8^ is 1.7% higher than that with an E_cr_ value of 1.2 × 10^8^. Due to premature brittle damage, the ceramic particles with a smaller E_cr_ (1.0 × 10^8^) experience stress relaxation and degradation in the driving force of crack evolution in the matrix region adjacent to the damaged particle, indicating that the stress level in the matrix is reduced for the ceramic with a small E_cr_ (1.0 × 10^8^). Therefore, the damage pixel with an E_cr_ of 1.0 × 10^8^ is 1.6% smaller than that with an E_cr_ (1.2 × 10^8^) in the austenitic matrix. Thus, the increased critical energy release rate of the ceramic particles causes a difference in the microstructure failure mechanism. In addition, the severe condition of the ceramic particles is detected at 20.3% global strain in the smaller E_cr_ (1.0 × 10^8^) scenario. In the austenitic matrix, the severe condition is observed in the larger E_cr_ (1.2 × 10^8^) scenario. It can be concluded that an increase in the strain release rate of the ceramic particles would result in severe damage to the matrix material. In this section, the qualitative and quantitative damage behaviors, such as propagation orientation and damage pixel in simulation, are compared.

## 4. Discussion

In this article, to develop and apply a dislocation-density-based crystal plasticity model with ductile damage criteria for the austenitic matrix and brittle damage criteria for ceramic particles, a DAMASK simulation is employed. The geometry model is imported to simulate an actual micrograph from an in situ test specimen. Finally, the global and local results of the simulation are determined and validated using experiments incorporating deformation and damage events. According to experimental findings, the damage mechanism is dominated by the interfacial cracking during the tensile test, as illustrated in Figure 4. The crack propagates along the interface, resulting in full debonding between the ceramic particle and matrix. However, the particle and matrix cracking that occur during the compressive test significantly dominate the damage mechanism of the TRIP steel reinforced by irregular ceramic particles [51]. Particularly, the propagation of the inner crack in particles is mainly along the loading axis and leads to the fracture of particles. The results reveal that the damage mechanism of the TRIP steel reinforced by ceramic particles is sensitive to loading type. Due to debonding, there is no impact of reinforcement in TRIP composites on deformation behavior, even without the ceramic particle cracking during the tensile test.

The comparison of the local results between the simulation and the experiment in Figure 5 shows that the strain distribution in the case of 2D-RVE is slightly different from that of the examined DIC. This difference is attributed to the dimensional difference between 2D and 3D. The simulation is considered 2D geometry, while the experiment is investigated in 3D specimens. In the 2D-RVE case [26,52], the strain concentration is observed in the specific matrix region. Meanwhile, near the 3D-RVE [53,54], the larger strain transfers to the ceramic/matrix interface. Thus, damage evolution differs in the case of 2D and 3D due to different strain distributions in 2D and 3D. According to the 3D simulation result of the evolution of the damage, the crack begins at the interface and propagates along the depth direction, which is similar to the experimental result.

Figure 9 depicts the damage pixel with different global strains from the experiment and two simulation results for comparing the effect of the energy release rates of ceramic particles by simulation and the experimental validation by qualitative and quantitative assessment.

The fourth step, which is indicated by a red dashed line in Figure 7, determines the damage pixel for the experiment. As shown in Figure 8, the damage pixel for the two simulation results is obtained by normalizing the grey pixel (damage) to the total pixel of the micrograph. In the qualitative damage figure stated in the top part, the E_cr_ at the right side or middle is 1.2 × 10^8^ and 1.0 × 10^8^ Jm^−2^ at a global strain of 20.3%, and the micrograph with the damage detected is on the left side. In the simulation figure, the grey color represents damage, the red color represents ceramic particles, and the blue color represents the austenite matrix. In the experiment figure, the yellow-colored damage pixel is detected by MATLAB, as mentioned in Section 3.3.1.

As per the qualitative result, for the simulation with critical plastic strain ductile damage criteria, the crack begins at the high plastic strain due to larger deformation in the austenite matrix and propagates at 45 degrees to the loading direction when the material exceeds the critical plastic strain [23]. However, as shown in Figure 9, a different damage distribution exists between the simulation and the experiment. The crack initiation is observed in the high plastic region in the simulation, while interfacial cracks are detected in the experimental observation. There are two primary reasons. The first reason is the aforementioned difference in dimensions between 2D and 3D [54,55]. The second reason is the absence of realistic grain orientation and Euler angles. In this study, three Euler angles were defined as 0 in the initial stage. As known, grain orientation has a significant impact on both global and local stress. 

The damage pixel exhibits an exponentially increasing trend in the quantitative result. It is observed that the damage value continually increases when the strain and energy exceed the critical value. In previous research [56], the theoretical interfacial debonding model for the particle-reinforced composite focused on the debonding area and the softening effect on the composites. The findings demonstrated that when the damaged area reaches the critical value, a visible reduction in stiffness occurs, implying that the stiffer reinforced particles are degraded. Meanwhile, the effect of reinforcement on the composite gradually diminished and degraded. In contrast to the predictions of numerical simulations, the damage evolution in experiments begins earlier. As per the damage results for the experiment, an immediate increase is observed at two intervals. This damage is considered an interfacial failure at a range of 2.8% to 9% along the interface between austenite and ceramic. Although the material can function at this stage, it is hazardous. At 20.3%, the composite exhibits a devastating catastrophe, indicating that the stress attains the ultimate strength and almost destroys the specimen. To accurately compare experiments and simulations, the relative error of the damage pixel is defined as follows:(3)$$relative\;error=\frac{N_{Sim.}-N_{Exp.}}{N_{Exp.}}\times 100\;\%$$
where $N_{Sim.}$ and $N_{Exp.}$ denote the damage pixel for the simulation and experiment method. The relative error of the smaller E_cr_ (1.0 × 10^8^) and larger E_cr_ (1.2 × 10^8^) cases at a global strain is 2.36% and 22.07%, respectively. As shown, the smaller E_cr_ (1.0 × 10^8^) case is more consistent with the experimental result. The absence of the 3D real microstructure and grain orientation is assumed to have caused a slight error in the qualitative deformation and damage result. Future developments involve acquiring and importing the realistic 3D RVE model and EBSD data into the material property. To understand the accuracy of the deformation and damage evolution, further comparison with experimental data and simulation results aided by 3D microstructure and EBSD is beneficial.

## 5. Conclusions

In this study, local deformation and the evolution of damage in a realistic microstructure in a TRIP steel matrix with ceramic particles were investigated. The effect of the energy release rate of the ceramic particles was specifically examined through simulation and experimental validation. A dislocation density plasticity crystal model with ductile damage for austenite and brittle damage for ceramic particles was implemented. Accordingly, two different energy release rates of the ceramic particles were determined by the damage behavior of the microstructure. In situ monotonic tensile loading was applied on the dogbone-shaped specimen and the global results were collected. The microstrain distribution and damage pixel were identified by digital image correlation (VEDDAC) and the image processing method (MATLAB). Based on the analysis of the experimental and simulation results, the investigation can be concluded as follows:When it is assumed that the critical plastic strain is 0.75 and the energy release rate is 1.0 × 10^8^ Jm^−2^, the global behavior of the stress and strain curve agrees well with the experiment and the simulation model.Through the MATLAB image processing function, the damage region is detected in four steps, and the damage pixel is quantitatively analyzed. At 20.3% global strain, the damage pixels in the fourth step (flood fill) and third step (free-hand ROI) are 4.9% and 2.8%, respectively, which indicates that the fourth step is 2.1% higher than the third step. The damage pixel in the fourth step can be regarded as a more realistic condition.The different energy release rates of the ceramic particles cause variations in the microstructure failure mechanism. This implies that the ceramic particles with severe damage are detected in the smaller E_cr_ (1.0 × 10^8^) case at 20.3% global strain. However, the austenitic matrix with severe damage is found in the larger E_cr_ (1.2 × 10^8^) case.Based on the quantitative damage result, at a global strain of 20.3%, the damage pixel of the E_cr_ (1.0 × 10^8^) case in ceramic particles is 1.7% larger than the E_cr_ (1.2 × 10^8^) case. As a consequence of premature brittle damage, the smaller E_cr_ (1.0 × 10^8^) case experiences stress relaxation and degradation of the driving force of crack evolution in the matrix region adjacent to the damaged particle. Conversely, the damage pixel of E_cr_ (1.0 × 10^8^) is 1.6% smaller than the E_cr_ (1.2 × 10^8^) case in the austenite matrix. Therefore, an increase in the strain release rate of the ceramic particles will result in severe damage to the matrix material. The damage pixel of the experiment, smaller E_cr_ (1.0 × 10^8^), and larger E_cr_ (1.2 × 10^8^) cases are 4.9%, 4.3%, and 5.1%, respectively. Furthermore, on a global strain of 20.3%, the relative errors between simulation and experimental validation of smaller E_cr_ (1.0 × 10^8^) and larger E_cr_ (1.2 × 10^8^) cases are –12.2% and 4%, respectively.It can be demonstrated that there is a slight difference in the qualitative damage and the local strain distribution between the crystal simulation and the experiment. The first difference is between the dimensions of the simulation (2D) and the experiment (3D). The absence of the initial grain orientation from EBSD is the second difference. Therefore, an initial grain orientation and a precise 3D microstructure in the composite material are required for further investigation.

## Figures and Tables

**Figure 1 materials-16-00134-f001:**
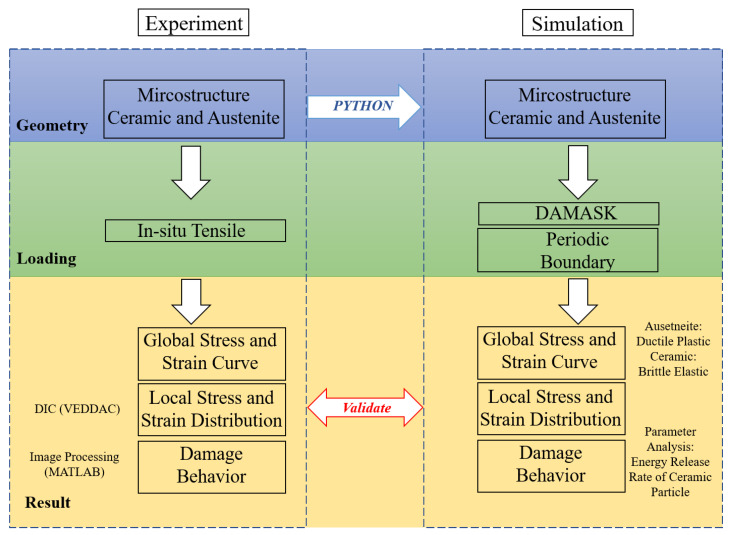
The outlined diagram represents the flow chart of the present work. The left-hand side represents the experimental process, and the right-hand side shows the simulation process. The initial SEM image is imported into the simulation using a Python script. In addition, the global, local, and damage analyses are compared with those of the experiment.

**Figure 2 materials-16-00134-f002:**
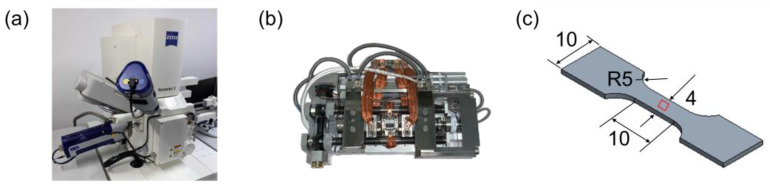
(**a**) ZEISS Gemini SEM 450 scanning electron microscope and (**b**) the in situ tensile instrument. (**c**) The geometry of the in situ samples, with a total length of 42 mm and a thickness of 1 mm, where all dimensions are in mm. The figures have been adapted from a previous publication [29] and are being reprinted with the permission of open access article.

**Figure 3 materials-16-00134-f003:**
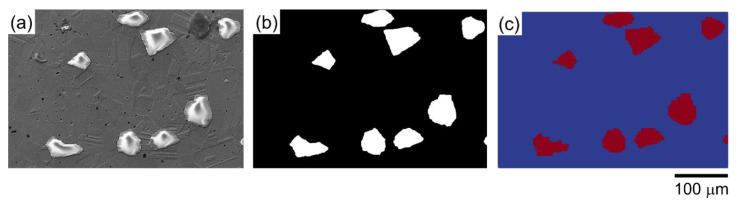
(**a**) SEM image obtained at the initial stage, (**b**) the revised 2D microstructure of ceramic particles (white) and austenite matrix (black), and (**c**) the detected 2D microstructure of ceramic particles (red) and austenite matrix (blue), identified by Python and displayed by ParaView. Figure 3a has been adapted from a previous publication [29] and is being reprinted with the permission of open access article.

**Figure 4 materials-16-00134-f004:**
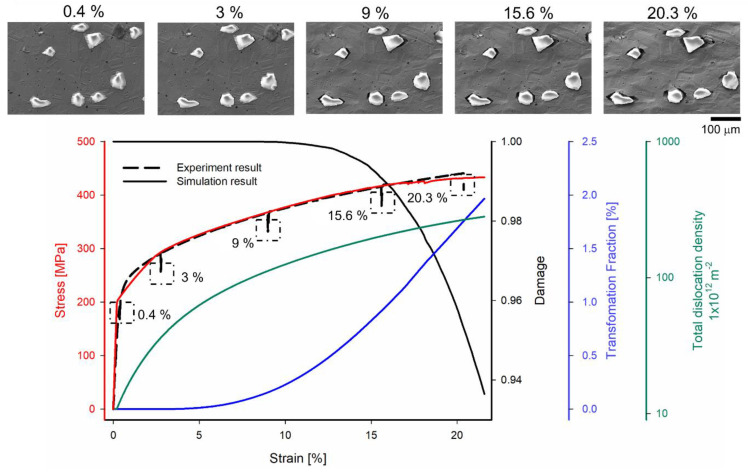
Average evolution of stress (experiment and simulation), phase transformation, damage degradation, and dislocation density in Mg-PSZ within the TRIP steel composite at different global true strains.

**Figure 5 materials-16-00134-f005:**
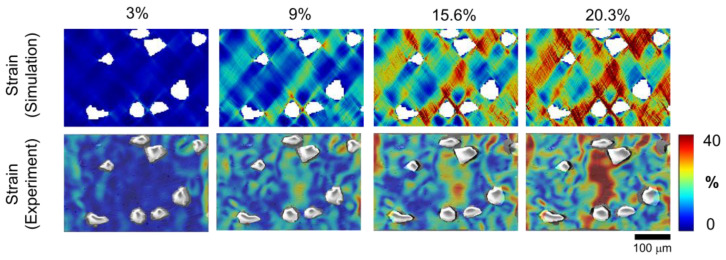
Local distribution of von Mises strains in the austenite matrix for simulation and experimental DIC at 3%, 9%, 15.6%, and 20.3% global strain conditions (from **left** to **right**).

**Figure 6 materials-16-00134-f006:**
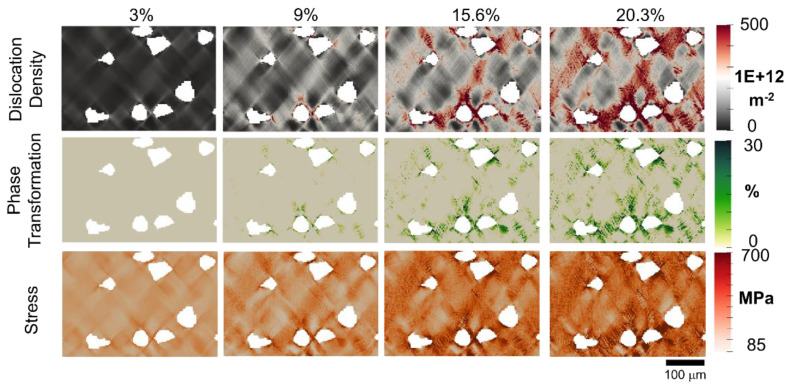
Simulation of the local distribution of dislocation density, phase transformation, and von Mises stress (from **top** to **bottom**) in the austenite matrix at global strains of 3%, 9%, 15.6%, and 20.3% (from **left** to **right**).

**Figure 7 materials-16-00134-f007:**
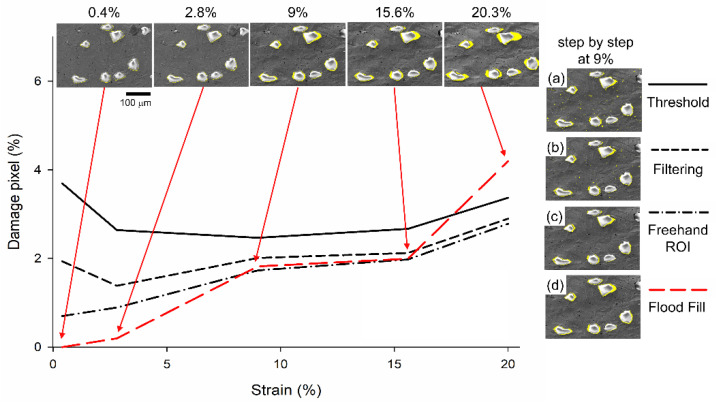
The four-step image detection process that includes (**a**) threshold, (**b**) filtering, (**c**) free-hand ROI, and (**d**) flood fill is displayed at 9% global deformation on the right-hand side. The quantitative comparison between the damage pixel and the global strain for four steps is presented on the left-hand side. On the upper side, the qualitative damage distribution is displayed through the fourth step.

**Figure 8 materials-16-00134-f008:**
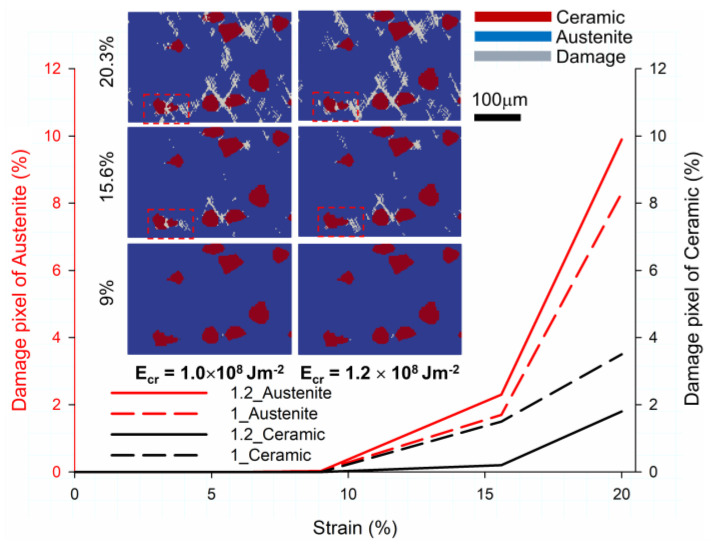
Simulation of the local damage distribution in the austenite matrix and the inclusion under global strain conditions of 9%, 15.6%, and 20% with different critical strain energies (E_cr_) of ceramic. For the qualitative damage figure, the E_cr_ on the left side is 1.0 × 10^8^ Jm^−2^ and on the right side is 1.2 × 10^8^ Jm^−2^.

**Figure 9 materials-16-00134-f009:**
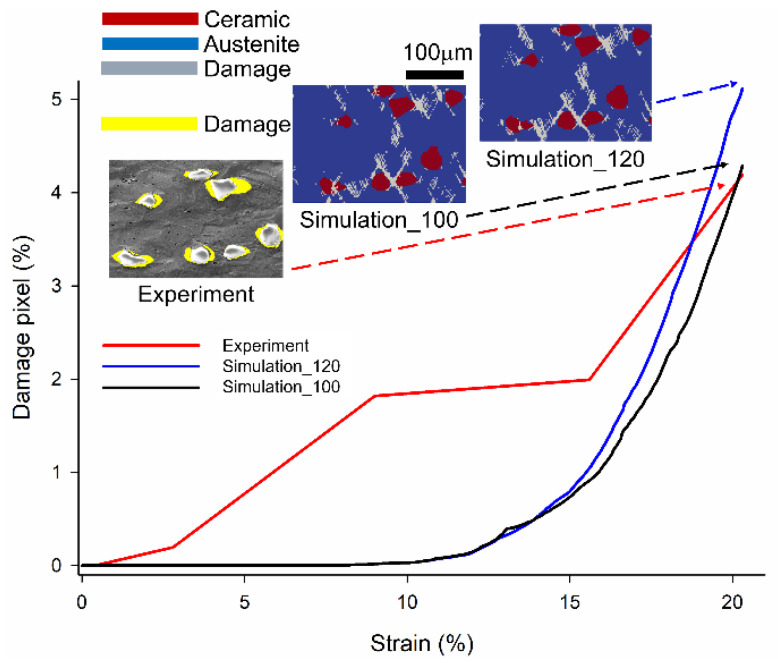
Global damage level and qualitative result with different global strain conditions of 9%, 15.6%, and 20% global strain for the experiment, and two simulation results with different critical energy release rates (E_cr_) of ceramic particles.

**Table 1 materials-16-00134-t001:** Chemical composition of steel and ZrO_2_ (in wt.-%). The table has been adapted from a previous publication [29] and are being reprinted with the permission of open access article.

Steel Alloy	C	Mn	Si	Cr	Ni	N	Fe
16–7–6	0.03	7.2	1.0	16.3	6.6	0.09	bal.
Ceramic	ZrO_2_	MgO	Na_2_O	CaO	TiO_2_	Fe_2_O_3_	SiO_2_
Mg-PSZ	94.14	2.82	0.1	0.15	0.13	0.13	0.41

**Table 2 materials-16-00134-t002:** Optimized constitutive model parameters for X3CrMnNi16–7–6 TRIP steel incorporated in the model during DAMASK simulations.

	Symbol	Description	Value	Unit	Ref.
Dislocation Slip Parameters	*b_s_*	Burgers vector of slip	2.56 × 10^−10^	m	[32]
	*ρ_e_*	Edge dislocation density	1.0 × 10^12^	m/m^3^	[41]
	*D_0_*	Self-diffusion coefficient for fcc Fe	4.0 × 10^−5^	m^2^/s	[32]
	*v_0_*	Dislocation glide velocity	1.0 × 10^−4^	m/s	[32]
	*q*	Bottom of the obstacle profile	1.0	-	[32]
	*p*	Top of the obstacle profile	1.15	-	[32]
	*Q_c_*	Activation energy for the climb	3.0 × 10^−19^	J	[32]
	*Q_s_*	Activation energy for glide	3.5 × 10^−19^	J	[32]
	*τ_sol_*	Solid solution strength	5 × 10^7^	Pa	
	*λ_slip_*	parameter controlling dislocation mean free path	55	-	[23]
	*d*	Average Grain size	2 × 10^−5^	m	[42]
Twinning Formation Parameters	*b_tw_*	Burgers vector of twin system	1.2 × 10^−10^	m	[32]
	*t_tw_*	Average twin thickness	5 × 10^−8^	m	[32]
	*V_cs_*	Cross-slip activation volume	1.67 × 10^−29^	m^3^	[32]
	*A*	Twinning transition profile width exponent	1.0	-	[23]
	*λ_sliptwin_*	Parameter controlling twin mean free path	5	-	[23]
	${\hat{\tau }}_{tw}$	Parameter controlling twin threshold stress	1.3	MPa	[23]
	*Γ* * _sf_ *	Stacking fault energy	10	mJ/m^2^	[43]
Martensite Transformation Parameters	*b_tr_*	Burgers vector of the trans system	1.47 × 10^−10^	m	[32]
	*t_tr_*	Average martensite thickness	5 × 10^−6^	m	[44]
	*V_cs_*	Cross-slip activation volume	1.67 × 10^−29^	m^3^	[32]
	*B*	Transformation transition profile width exponent	3.0	-	[23]
	*λ_sliptran_*	Parameter controlling trans. mean free path	10	-	[23]
	${\hat{\tau }}_{tr}$	Parameter controlling trans threshold stress	0.5	MPa	[23]
	*h*	Height of the hcp nucleus	1.06 × 10^−9^		[45]
	*ΔG* * ^γ^ * * ^→^ * * ^ε^ *	Change in Gibbs free energy	−2.54 × 10^7^	J/m^3^	[46]

**Table 3 materials-16-00134-t003:** Single-crystal elastic constants for austenite, martensite, and ceramic were incorporated into the model during simulations. The table has been adapted from a previous publication [14] and is being reprinted with the permission of MDPI.

Austenite	Martensite	Ceramic	Unit
C_11_ = 175.0	C_11_ = 191.0	C_11_ = 191.0	GPa
C_12_ = 115.0	C_12_ = 80.0	C_12_ = 80.0	GPa
C_44_ = 135.0	C_13_ = 40.0	C_44_ = 40.0	GPa
	C_33_ = 315.0		GPa
	C_44_ = 40.5		GPa

**Table 4 materials-16-00134-t004:** Physical and fitting brittle damage parameter values used for ceramic particles. The table has been adapted from a previous publication [23] and is being reprinted with the permission of MDPI.

Parameter Definition	Property	Value	Unit
Characteristic length	l_0_	1.0	μm
Damage mobility	M	0.001	-
Damage diffusion	D	1.0	-

**Table 5 materials-16-00134-t005:** Values of the physical and ductile damage parameters used for austenite steel. The table has been adapted from a previous publication [23] and is being reprinted with the permission of MDPI.

Parameter Definition	Property	Value	Unit
Critical plastic strain	ε_p,crit_	0.75	-
Characteristic length	l_0_	1.0	μm
Damage mobility	M	0.001	-
Damage diffusion	D	1.0	-
Damage rate sensitivity	P	35	-

## Data Availability

The simulation data are not publicly available but can be shared upon request.

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
