# Peer review of "The Effect of the Energy Release Rate on the Local Damage Evolution in TRIP Steel Composite Reinforced with Zirconia Particles"

_materials, 2022, doi:10.3390/ma16010134_

Round 1
Reviewer 1 Report
After a thorough analysis of the article submitted by the authors to be published in the MDPI Materials journal, I totally agree with its publication in its current form, since I have found this article very professional realized and presented in a very systematic way by the authors.
Objectives of the research are clearly stated in the paper, the research methods for manufacturing and testing of the samples are very well provided in the text with really nice scientific soundness. Results are very well provided both in graphical mode accompanied with very detailed explanations. I really appreciated how things are argued in the text and how comparison of the results are provided both by comparing the experimental results with the results reached by the simulations, but also with references to the work done by other researchers in the domain (in the discussion section). Modern software programs are used like Pyton in the research. Also references are really actual and very consistent. English style language (speling and grammar) is super good and technical terms and expressions I have found in the text are excellent. Plagiarism checking provided good scores after a very thorough checking as well (IThenticate program I have used it for this type of checking).
In conclusion of my report as I have stated from the beginning, I have found this paper excellent and professionally presented by the authors and due to all reasons I have provided in detail in the paragraphs presented above, I consider that this paper is really one plus for the MDPI jMaterials ournal and therefore I definitely recommend this paper to be accepted as it is in the MDPI Materials journal.
Congratulations to all authors for very good results that were provided in a very well-written scientific article aimed to be published in the MDPI Materials journal this year!
Author Response
We are thankful to the reviewers for taking the time to review the article and providing valuable feedback, which helped us improve the article's clarity and outlook. All the reviewer comments have been appropriately addressed in the main article body, and the modified text is highlighted yellow for easier comparison. The article was read thoroughly to check for any grammatical or punctuation mistakes and improvement in the clarity of language professionally. The article's methods, results, and conclusion sections have been coherently revised to improve understanding and clarity.
A point-by-point response to each reviewer's comment with reason and subsequent modification in the article body is provided in the attached PDF document.

Reviewer 2 Report
The material under investigation in this work is zirconia toughened steel. The strengthening effect is attributed to the occurrence of phase transformation triggered by the stress field around existing crack tips. While this phenomenon has widely been investigated in the literature the authors focus on the impact of the energy release rate of the ceramic particles as a novel aspect by means of a numerical study. To this end they take advantage of existing software that has been published and used in previous works. That software takes care of phase transformation in combination with crystal plasticity thereby enabling to study the onset of damage expressed in terms of an equivalent strain measure which is then compared to the results of concomitant experiments.
1. All in all the focus on studying the influence of the energy release rate of the ceramics as the novel aspect of the work is quite narrow for a scientific publication and the trends are qualitatively predictable even without further analysis. At this point quantitative statements are not provided due to the 2D nature of the analysis. The reviewer suggests to repeat the analysis using a 3D model (as long as this is computationally feasible). Of course, DIC data are only available on the specimen’s surface so a real microstructure simulation is not possible in 3D. However, for the point the authors are making in their work an artificial 3D microstructure still allows a statistical comparison with experiments, such as e.g. in Figs. 7-9.
2. There are also some language issues. In the introduction the use of the past tense (starting with line 49) is not the best choice. Many expressions are not suitable or a word seems to be missing (e.g. line 152, 296, 412). The manuscript should be thoroughly proofread before resubmission.
3. The black line in Fig. 4 representing the damage starts at 1 and then decreases? How is this possible? The expectation is that it starts at zero and then increases. An clarification should be provided in the manuscript.
This reviewer does not recommend publication of the paper in its current form. It can be made publishable, though, by addressing the issues mentioned above. The recommendation flag is set to “major revision”.
Author Response

(The authors gave the same response as above.)

Reviewer 3 Report
This paper performed to Effect of Energy Release Rate on the Transformation-Induced Plasticity Steel Composite Reinforced with 5 Vol-% Ceramic Particles: Numerical and Experimental Study. The article is, in general, well written but there are some issues that authors should consider to revise in order to improve its quality. Some comments were done in this way:
· The title of the article is long and should be shortened to be novelty.
· Abstract should be expanded sentences related to the results. The results of the study should be given as numerical percentages.
· The paper should be also supported by a literature search including relevant and recent papers. Only the most relevant and up-to-date articles on the study should be given.
· Let's fix grammatical errors throughout the article.
· Give the finite element analysis parameters in a table.
· Give the parameters of the material model used in the finite element analysis in a separate table.
· The article should be edited completely according to the journal writing guide.
· Throughout the article, the words table and figure should start with capital letters (Table, Figure).
· Fractions should be given with dots throughout the article, including figures and tables.
· Expand the discussion section using current literature.
· Conclusions should be written in more detail adding numeric data.
Author Response

(The authors gave the same response as above.)

Round 2
Reviewer 2 Report
The authors responded satisfactorily to the issues mentioned in the first review report. The paper can now be published.
Reviewer 3 Report
---